# Scalable Gromov-Wasserstein Learning for Graph Partitioning and Matching

**Hongteng Xu**[1,2]     **Dixin Luo**[2]     **Lawrence Carin**[2]
[1]Infinia ML Inc.     [2]Duke University
{hongteng.xu, dixin.luo, lcarin}@duke.edu

## Abstract

We propose a scalable Gromov-Wasserstein learning (S-GWL) method and establish a novel and theoretically-supported paradigm for large-scale graph analysis. The proposed method is based on the fact that Gromov-Wasserstein discrepancy is a pseudometric on graphs. Given two graphs, the optimal transport associated with their Gromov-Wasserstein discrepancy provides the correspondence between their nodes and achieves graph matching. When one of the graphs has isolated but self-connected nodes (*i.e.*, a disconnected graph), the optimal transport indicates the clustering structure of the other graph and achieves graph partitioning. Using this concept, we extend our method to multi-graph partitioning and matching by learning a Gromov-Wasserstein barycenter graph for multiple observed graphs; the barycenter graph plays the role of the disconnected graph, and since it is learned, so is the clustering. Our method combines a recursive $K$-partition mechanism with a regularized proximal gradient algorithm, whose time complexity is $\mathcal{O}(K(E+V)\log_K V)$ for graphs with $V$ nodes and $E$ edges. To our knowledge, our method is the first attempt to make Gromov-Wasserstein discrepancy applicable to large-scale graph analysis and unify graph partitioning and matching into the same framework. It outperforms state-of-the-art graph partitioning and matching methods, achieving a trade-off between accuracy and efficiency.

## 1   Introduction

Gromov-Wasserstein distance [42, 29] was originally designed for metric-measure spaces, which can measure distances between distributions in a relational way, deriving an optimal transport between the samples in distinct spaces. Recently, the work in [11] proved that this distance can be extended to *Gromov-Wasserstein discrepancy* (GW discrepancy) [37], which defines a pseudometric for graphs. Accordingly, the optimal transport between two graphs indicates the correspondence between their nodes. This work theoretically supports the applications of GW discrepancy to structural data analysis, *e.g.*, 2D/3D object matching [30, 28, 8], molecule analysis [43, 44], network alignment [49], etc. Unfortunately, although GW discrepancy-based methods are attractive theoretically, they are often inapplicable to large-scale graphs, because of high computational complexity. Additionally, these methods are designed for two-graph matching, ignoring the potential of GW discrepancy to other tasks, like graph partitioning and multi-graph matching. As a result, the partitioning and the matching of large-scale graphs still typically rely on heuristic methods [16, 12, 45, 27], whose performance is often sub-optimal, especially in noisy cases.

Focusing on the issues above, we design a scalable Gromov-Wasserstein learning (S-GWL) method and establish a new and unified paradigm for large-scale graph partitioning and matching. As illustrated in Figure 1(a), given two graphs, the optimal transport associated with their Gromov-Wasserstein discrepancy provides the correspondence between their nodes. Similarly, graph partitioning corresponds to calculating the Gromov-Wasserstein discrepancy between an observed graph and a disconnected graph, as shown in Figure 1(b). The optimal transport connects each node of the ob-

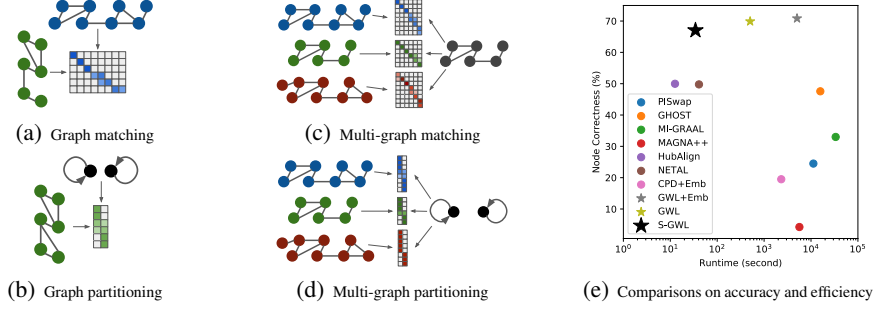

(a) Graph matching    (c) Multi-graph matching    (e) Comparisons on accuracy and efficiency

(b) Graph partitioning    (d) Multi-graph partitioning

Figure 1: (a)-(d) Illustrations of graph partitioning and matching in the GWL framework. (c, d) The barycenter graph in black and its optimal transports to observed graphs are learned jointly. (d) When the barycenter graph is initialized as a graph with few isolated nodes, the optimal transports indicate aligned partitions of observed graph. (e) We test various graph matching methods in 10 trials on an Intel i7 CPU. In each trial, the source graph has 2,000 nodes and the target graph has 100 more noisy nodes and corresponding edges. The graphs yield either Gaussian partition model [7] or Barabási-Albert model [4]. The GWL-based methods ('★') obtains higher node correctness than other baselines ('●'), and our S-GWL (big '★') achieves a trade-off on accuracy and efficiency.

served graph with an isolated node of the disconnected graph, yielding a partitioning. In Figures 1(c) and 1(d), taking advantage of the Gromov-Wasserstein barycenter in [37], we achieve multi-graph matching and partitioning by learning a "barycenter graph". For arbitrary two or more graphs, the correspondence (or the clustering structure) among their nodes can be established indirectly through their optimal transports to the barycenter graph.

The four tasks in Figures 1(a)-1(d) are explicitly unified in our Gromov-Wasserstein learning (GWL) framework, which corresponds to the same GW discrepancy-based optimization problem. To improve its scalability, we introduce a recursive mechanism to the GWL framework, which recursively applies $K$-way partitioning to decompose large graphs into a set of aligned sub-graph pairs, and then matches each pair of sub-graphs. When calculating GW discrepancy, we design a regularized proximal gradient method, that considers the prior information of nodes and performs updates by solving a series of convex sub-problems. The sparsity of edges further helps us reduce computations. These acceleration strategies yield our S-GWL method: for graphs with $V$ nodes and $E$ edges, its time complexity is $\mathcal{O}(K(E+V)\log_K V)$ and memory complexity is $\mathcal{O}(E+VK)$. To our knowledge, our S-GWL is the first to make GW discrepancy applicable to large-scale graph analysis. Figure 1(e) illustrates the effectiveness of S-GWL on graph matching, with more results presented in Section 5.

## 2 Graph Analysis Based on Gromov-Wasserstein Learning

Denote a *measure graph* as $G(\mathcal{V}, \boldsymbol{C}, \boldsymbol{\mu})$, where $\mathcal{V} = \{v_i\}_{i=1}^{|\mathcal{V}|}$ is the set of nodes, $\boldsymbol{C} = [c_{ij}] \in \mathbb{R}^{|\mathcal{V}| \times |\mathcal{V}|}$ is the adjacency matrix, and $\boldsymbol{\mu} = [\mu_i] \in \Sigma^{|\mathcal{V}|}$ is a Borel probability measure defined on $\mathcal{V}$. The adjacency matrix is continuous for weighted graph while binary for unweighted graph. In practice, $\boldsymbol{\mu}$ is an empirical distribution of nodes, which can be estimated by a function of node degree. A $K$-way graph partitioning aims to decompose a graph $G$ into $K$ sub-graphs by clustering its nodes, *i.e.*, $\{G_k = G(\mathcal{V}_k, \boldsymbol{C}_k, \boldsymbol{\mu}_k)\}_{k=1}^K$, where $\cup_k \mathcal{V}_k = \mathcal{V}$ and $\mathcal{V}_k \cap \mathcal{V}_{k'} = \emptyset$ for $k \neq k'$. Given two graphs $G_s$ and $G_t$, graph matching aims to find a correspondence between their nodes, *i.e.*, $\pi : \mathcal{V}_s \mapsto \mathcal{V}_t$. Many real-world networks are modeled using graph theory, and graph partitioning and matching are important for community detection [21, 16] and network alignment [39, 40, 54], respectively. In this section, we propose a Gromov-Wasserstein learning framework to unify these two problems.

### 2.1 Gromov-Wasserstein discrepancy between graphs

Our GWL framework is based on a pseudometric on graphs called Gromov-Wasserstein discrepancy:

**Definition 2.1** ([11]). *Denote the collection of measure graphs as $\mathcal{G}$. For each $p \in [1, \infty]$ and each $G_s, G_t \in \mathcal{G}$, the Gromov-Wasserstein discrepancy between $G_s$ and $G_t$ is*

$$d_{gw}(G_s, G_t) := \min_{\boldsymbol{T} \in \Pi(\boldsymbol{\mu}_s, \boldsymbol{\mu}_t)} \Big( \sum_{i,j \in \mathcal{V}_s} \sum_{i',j' \in \mathcal{V}_t} |c_{ij}^s - c_{i'j'}^t|^p T_{ii'} T_{jj'} \Big)^{\frac{1}{p}}, \qquad (1)$$

*where $\Pi(\boldsymbol{\mu}_s, \boldsymbol{\mu}_t) = \{\boldsymbol{T} \geq 0 | \boldsymbol{T} \boldsymbol{1}_{|\mathcal{V}_t|} = \boldsymbol{\mu}_s, \boldsymbol{T}^\top \boldsymbol{1}_{|\mathcal{V}_s|} = \boldsymbol{\mu}_t\}$.*

GW discrepancy compares graphs in a relational way, measuring how the edges in a graph compare to those in the other graph. It is a natural extension of the Gromov-Wasserstein distance defined for metric-measure spaces [29]. We refer the reader to [29, 11, 36] for mathematical foundations.

**Graph matching** According to the definition, GW discrepancy measures the distance between two graphs, and the optimal transport $\boldsymbol{T} = [T_{ij}] \in \Pi(\boldsymbol{\mu}_s, \boldsymbol{\mu}_t)$ is a joint distribution of the graphs' nodes: $T_{ij}$ indicates the probability that the node $v_i^s \in \mathcal{V}_s$ corresponds to the node $v_j^t \in \mathcal{V}_t$. As shown in Figure 1(a), the optimal transport achieves an assignment of the source nodes to the target ones.

**Graph partitioning** Besides graph matching, this paradigm is also suitable for graph partitioning. Recall that most existing graph partitioning methods obey the modularity maximization principle [16, 12]: for each partitioned sub-graph, its internal edges should be dense, while its external edges connecting with other sub-graphs should be sparse. This principle implies that if we treat each sub-graph as a "super node" [21, 47, 34], an ideal partitioning should correspond to a disconnected graph with $K$ isolated, but self-connected super nodes. Therefore, we achieve $K$-way partitioning by calculating the GW discrepancy between the observed graph $G$ and a disconnected graph, *i.e.*, $d_{gw}(G, G_{dc})$, where $G_{dc} = G(\mathcal{V}_{dc}, \text{diag}(\boldsymbol{\mu}_{dc}), \boldsymbol{\mu}_{dc})$. $|\mathcal{V}_{dc}| = K$. $\boldsymbol{\mu}_{dc} \in \Sigma^K$ is a node distribution, whose derivation is in Appendix A.1. $\text{diag}(\boldsymbol{\mu}_{dc})$ is the adjacency matrix of $G_{dc}$. As shown in Figure 1(b), the optimal transport is a $|\mathcal{V}| \times K$ matrix. The maximum in each row of the matrix indicates the cluster of a node.

## 2.2 Gromov-Wasserstein barycenter graph for analysis of multiple graphs

**Multi-graph matching** Distinct from most graph matching methods [17, 13, 39, 14], which mainly focus on two-graph matching, our GWL framework can be readily extended to multi-graph cases, by introducing the Gromov-Wasserstein barycenter (GWB) proposed in [37]. Given a set of graphs $\{G_m\}_{m=1}^M$, their $p$-order Gromov-Wasserstein barycenter is a *barycenter graph* defined as

$$G(\bar{\mathcal{V}}, \bar{\boldsymbol{C}}, \bar{\boldsymbol{\mu}}) := \arg\min_{\bar{G}} \sum_{m=1}^M \omega_m d_{gw}^p(G_m, \bar{G}), \tag{2}$$

where $\boldsymbol{\omega} = [\omega_m] \in \Sigma^M$ contains predefined weights, and $\bar{G} = G(\bar{\mathcal{V}}, \bar{\boldsymbol{C}} \in \mathbb{R}^{|\bar{\mathcal{V}}| \times |\bar{\mathcal{V}}|}, \bar{\boldsymbol{\mu}} \in \Sigma^{|\bar{\mathcal{V}}|})$ is the barycenter graph with a predefined number of nodes. The barycenter graph minimizes the weighted average of its GW discrepancy to observed graphs. It is an average of the observed graphs aligned by their optimal transports. The matrix $\bar{\boldsymbol{C}}$ is a "soft" adjacency matrix of the barycenter. Its elements reflect the confidence of the edges between the corresponding nodes in $\bar{\mathcal{V}}$. As shown in Figure 1(c), the barycenter graph works as a "reference" connecting with the observed graphs. For each node in the barycenter graph, we can find its matched nodes in different graphs with the help of the corresponding optimal transport. These matched nodes construct a node set, and two arbitrary nodes in the set are a correspondence. The collection of all the node sets achieves multi-graph matching.

**Multi-graph partitioning** We can also use the barycenter graph to achieve multi-graph partitioning, with the *learned* barycenter graph playing the role of the aforementioned disconnected graph. Given two or more graphs, whose nodes may have unobserved correspondences, existing partitioning methods [21, 16, 12, 6, 34] only partition them independently because they are designed for clustering nodes in a single graph. As a result, the first cluster of a graph may correspond to the second cluster of another graph. Without the correspondence between clusters, we cannot reduce the search space in matching tasks. Although this correspondence can be estimated by matching two coarse graphs that treat the clusters as their nodes, this strategy not only introduces additional computations but also leads to more uncertainty on matching, because different graphs are partitioned independently without leveraging structural information from each other. By learning a barycenter graph for multiple graphs, we can partition them and align their clusters simultaneously. As shown in Figure 1(d), when applying $K$-way multi-graph partitioning, we initialize a disconnected graph with $K$ isolated nodes as the barycenter graph, and then learn it by $\min_{\bar{G}} \sum_{m=1}^M \omega_m d_{gw}^p(G_m, \bar{G})$. For each node of the barycenter graph, its matched nodes in each observed graph belong to the same cluster.

# 3 Scalable Gromov-Wasserstein Learning

Based on Gromov-Wasserstein discrepancy and the barycenter graph, we have established a GWL framework for graph partitioning and matching. To make this framework scalable to large graphs, we propose a regularized proximal gradient method to calculate GW discrepancy and integrate multiple acceleration strategies to greatly reduce the computational complexity of GWL.

## 3.1 Regularized proximal gradient method

Inspired by the work in [48, 49], we calculate the GW discrepancy in (1) based on a proximal gradient method, which decomposes a complicated non-convex optimization problem into a series of convex sub-problems. For simplicity, we set $p = 2$ in (1, 2). Given two graphs $G_s = G(\mathcal{V}_s, \boldsymbol{C}_s, \boldsymbol{\mu}_s)$ and

$G_t = G(\mathcal{V}_t, \boldsymbol{C}_t, \boldsymbol{\mu}_t)$, in the $n$-th iteration, we update the current optimal transport $\boldsymbol{T}^{(n)}$ by calculating $d_{gw}^2(G_s, G_t)$:

$$
\begin{aligned}
\boldsymbol{T}^{(n+1)} &= \arg\min_{\boldsymbol{T}\in\Pi(\boldsymbol{\mu}_s,\boldsymbol{\mu}_t)} \sum_{i,j\in\mathcal{V}_s} \sum_{i',j'\in\mathcal{V}_t} |c_{ij}^s - c_{i'j'}^t|^2 T_{ii'}^{(n)} T_{jj'}^{(n)} + \gamma\mathrm{KL}(\boldsymbol{T}\|\boldsymbol{T}^{(n)}) \\
&= \arg\min_{\boldsymbol{T}\in\Pi(\boldsymbol{\mu}_s,\boldsymbol{\mu}_t)} \langle \boldsymbol{L}(\boldsymbol{C}_s, \boldsymbol{C}_t, \boldsymbol{T}^{(n)}), \boldsymbol{T}\rangle + \gamma\mathrm{KL}(\boldsymbol{T}\|\boldsymbol{T}^{(n)}).
\end{aligned}
\tag{3}
$$

Here, $\boldsymbol{L}(\boldsymbol{C}_s, \boldsymbol{C}_t, \boldsymbol{T}) = \boldsymbol{C}_s\boldsymbol{\mu}_s\mathbf{1}_{|\mathcal{V}_t|}^\top + \mathbf{1}_{|\mathcal{V}_s|}\boldsymbol{\mu}_t^\top\boldsymbol{C}_t^\top - 2\boldsymbol{C}_s\boldsymbol{T}\boldsymbol{C}_t^\top$, derived based on [37], and $\langle\cdot,\cdot\rangle$ represents the inner product of two matrices. The Kullback-Leibler (KL) divergence, $i.e.$, $\mathrm{KL}(\boldsymbol{T}\|\boldsymbol{T}^{(n)}) = \sum_{ij} T_{ij}\log(T_{ij}/T_{ij}^{(n)}) - T_{ij} + T_{ij}^{(n)}$, is added as the proximal term. We can solve (3) via the Sinkhorn-Knopp algorithm [41, 15] with nearly-linear convergence [1]. As demonstrated in [49], the global convergence of this proximal gradient method is guaranteed, so repeating (3) leads to a stable optimal transport, denoted as $\widehat{\boldsymbol{T}}$. Additionally, this method is robust to hyperparameter $\gamma$, achieving better convergence and numerical stability than the entropy-based method in [37].

Learning the barycenter graph is also based on the proximal gradient method. Given $M$ graphs, we estimate their barycenter graph via alternating optimization. In the $n$-th iteration, given the previous barycenter graph $\bar{G}^{(n)} = G(\bar{\mathcal{V}}, \bar{\boldsymbol{C}}^{(n)}, \bar{\boldsymbol{\mu}})$, we update $M$ optimal transports via solving (3). Given the updated optimal transports $\{\boldsymbol{T}_m^{(n+1)}\}_{m=1}^M$, we update the adjacency matrix of the barycenter graph by

$$
\bar{\boldsymbol{C}}^{(n+1)} = \frac{1}{\bar{\boldsymbol{\mu}}\bar{\boldsymbol{\mu}}^\top} \sum_m \omega_m (\boldsymbol{T}_m^{(n+1)})^\top \boldsymbol{C}_m \boldsymbol{T}_m^{(n+1)}.
\tag{4}
$$

The weights $\boldsymbol{\omega}$, the number of the nodes $|\bar{\mathcal{V}}|$ and the node distribution $\bar{\boldsymbol{\mu}}$ are predefined.

Different from the work in [49, 37], we use the following initialization strategies to achieve a regularized proximal gradient method and estimate optimal transports with few iterations.

**Node distributions** We estimate the node distribution $\boldsymbol{\mu}$ of a graph empirically by a function of node degree, which reflects the local topology of nodes, $e.g.$, the density of neighbors. In particular, for a graph with $|\mathcal{V}|$ nodes, we first calculate a vector of node degree, $i.e.$, $\boldsymbol{n} = [n_i] \in \mathbb{Z}^{|\mathcal{V}|}$, where $n_i$ is the number of neighbors of the $i$-th node. Then, we estimate the node distribution $\boldsymbol{\mu}$ as

$$
\boldsymbol{\mu} = \tilde{\boldsymbol{\mu}}/\|\tilde{\boldsymbol{\mu}}\|_1, \quad \tilde{\boldsymbol{\mu}} = (\boldsymbol{n} + a)^b.
\tag{5}
$$

where $a \geq 0$ and $b \geq 0$ are the hyperparameters controlling the shape of the distribution. For the graphs with isolated nodes, whose $n_i$'s are zeros, we set $a > 0$ to avoid numerical issues when solving (3). For the graphs whose nodes obey to power-law distributions, $i.e.$, Barabási-Albert graphs, we set $b \in [0, 1)$ to balance the probabilities of different nodes. This function generalizes the empirical settings used in other methods: when $a = 0$ and $b = 1$, we derive the distribution based on the normalized node degree used in [49]; when $b = 0$, we assume the distribution is uniform as the work in [37, 44] does. We find that the node distributions have a huge influence on the stability and the performance of our learning algorithms, which will be discussed in the following sections.

**Optimal transports** For graph analysis, we can leverage prior knowledge to get a better regularization of optimal transport. Generally, the nodes with similar local topology should be matched with a high probability. Therefore, given two node distributions $\boldsymbol{\mu}_s$ and $\boldsymbol{\mu}_t$, we construct a node-based cost matrix $\boldsymbol{C}_{\text{node}} \in \mathbb{R}^{|\mathcal{V}_s|\times|\mathcal{V}_t|}$, whose element is $c_{ij} = |\mu_i^s - \mu_j^t|$, and add a regularization term $\langle \boldsymbol{C}_{\text{node}}, \boldsymbol{T}^{(n)}\rangle$ to (3). As a result, in the learning phase, we replace the $\boldsymbol{L}(\boldsymbol{C}_s, \boldsymbol{C}_t, \boldsymbol{T}^{(n)})$ in (3) with $\boldsymbol{L}(\boldsymbol{C}_s, \boldsymbol{C}_t, \boldsymbol{T}^{(n)}) + \tau\boldsymbol{C}_{\text{node}}$, where $\tau$ controls the significance of $\boldsymbol{C}_{\text{node}}$. Introducing the proposed regularizer helps us measure the similarity between nodes directly, which extends our GW discrepancy to the fused GW discrepancy in [44, 43]. In such a situation, the main difference here is that we use the proximal gradient method to calculate the discrepancy, rather than the conditional gradient method in [43].

**Barycenter graphs** When learning GWB, the work in [37] fixed the node distribution to be uniform. In practice, however, both the node distribution of the barycenter graph and its optimal transports to observed graphs are unknown. In such a situation, we need to first estimate the node distribution $\bar{\boldsymbol{\mu}} = [\bar{\mu}_1, ..., \bar{\mu}_{|\bar{\mathcal{V}}|}]$. Without loss of generality, we assume that the node distribution of the barycenter graph is sorted, $i.e.$, $\bar{\mu}_1 \geq ... \geq \bar{\mu}_{|\bar{\mathcal{V}}|}$. We estimate the node distribution via the weighted average of the sorted and re-sampled node distributions of observed graphs:

$$
\bar{\boldsymbol{\mu}} = \sum_{m=1}^M \omega_m \text{interpolate}_{|\bar{\mathcal{V}}|}(\text{sort}(\boldsymbol{\mu}_m)),
\tag{6}
$$

| **Algorithm 1** ProxGrad($G_s, G_t, \gamma$) | **Algorithm 2** GWB($\{G_m\}_{m=1}^M, \gamma, |\bar{\mathcal{V}}|, \boldsymbol{\omega}$) |
|---|---|
| 1: Set $n = 0$, $\boldsymbol{a} = \boldsymbol{\mu}_s$. | 1: Set $n = 0$. |
| 2: Calculate $\boldsymbol{C}_{\text{node}}$ with $c_{ij} = |\mu_i^s - \mu_j^t|$. | 2: Initialize $\bar{\boldsymbol{\mu}}$ via (6). $\bar{\boldsymbol{C}}^{(n)} = \text{diag}(\bar{\boldsymbol{\mu}})$. |
| 3: Initialize $\boldsymbol{T}^{(n)} = \boldsymbol{\mu}_s \boldsymbol{\mu}_t^\top$. | 3: **While** not converge |
| 4: **While** not converge | 4:    **For** $m = 1, ..., M$ |
| 5:   $\boldsymbol{G} = e^{-(\boldsymbol{C}_{\text{node}} + \boldsymbol{L}(\boldsymbol{C}_s, \boldsymbol{C}_t, \boldsymbol{T}^{(n)}))/\gamma} \odot \boldsymbol{T}^{(n)}$. | 5:      $\boldsymbol{T}_m^{(n+1)} = \text{ProxGrad}(G_m, \bar{G}^{(n)}, \gamma)$. |
| 6:   $\boldsymbol{b} = \boldsymbol{\mu}_t/(\boldsymbol{G}^\top \boldsymbol{a})$, and $\boldsymbol{a} = \boldsymbol{\mu}_s/(\boldsymbol{G}\boldsymbol{b})$. | 6:    Calculate $\bar{\boldsymbol{C}}^{(n+1)}$ via (4). |
| 7:   $\boldsymbol{T}^{(n+1)} = \text{diag}(\boldsymbol{a})\boldsymbol{G}\text{diag}(\boldsymbol{b})$, then $n = n+1$. | 7:    $n = n + 1$. |
| 8: **Output:** $\widehat{\boldsymbol{T}} = \boldsymbol{T}^{(n)}$. | 8: **Output:** $\widehat{\boldsymbol{T}}_m = \boldsymbol{T}_m^{(n)}$ for $m = 1, .., M$. |

where sort($\cdot$) sorts the elements of the input vector in descending order, and interpolate$_{|\bar{\mathcal{V}}|}(\cdot)$ samples $|\bar{\mathcal{V}}|$ values from the input vector via bilinear interpolation. Given the node distribution, we initialize the optimal transports via the method mentioned above.

Algorithms 1 and 2 show the details of our method, where "$\odot$" and "$\cdot/\cdot$" represent elementwise multiplication and division, respectively. The GWL framework for the tasks in Figures 1(a)-1(d) are implemented based on these two algorithms, with details in Appendix A.1.

### 3.2   A recursive $K$-partition mechanism for large-scale graph matching

Assume that the observed graphs have comparable size, whose number of nodes and edges are denoted as $V$ and $E$, respectively. When using the proximal gradient method directly to calculate the GW discrepancy between two graphs, the time complexity, in the worst case, is $\mathcal{O}(V^3)$ because the $\boldsymbol{L}(\boldsymbol{C}_s, \boldsymbol{C}_t, \boldsymbol{T}^{(n)})$ in (3) involves $\boldsymbol{C}_s \boldsymbol{T} \boldsymbol{C}_t^\top$. Even if we consider the sparsity of edges and implement sparse matrix multiplications, the time complexity is still as high as $\mathcal{O}(EV)$.

To improve the scalability of our GWL framework, we introduce a recursive $K$-partition mechanism, recursively decomposing observed large graphs to a set of aligned small graphs. As shown in Figure 2(a), given two graphs, we first calculate their barycenter graph (with $K$ nodes) and achieve their joint $K$-way partitioning. For each node of the barycenter graph, the corresponding sub-graphs extracted from the observed two graphs construct an aligned sub-graph pair, shown as the dotted frames connected with grey circles in Figure 2(a). For each aligned sub-graph pair, we further calculate its barycenter graph and decompose the pair into more and smaller sub-graph pairs. Repeating the above step, we finally calculate the GW discrepancy between the sub-graphs in each pair, and find the correspondence between their nodes. Note that this recursive mechanism is also applicable to multi-graph matching: for multiple graphs, in the final step we calculate the GWB among the sub-graphs in each set. The details of our S-GWL method are provided in Appendix A.2.

**Complexity analysis** In Table 1, we compare the time and memory complexity of our S-GWL method with other matching methods. The Hungarian algorithm [24] has time complexity $\mathcal{O}(V^3)$ [17, 33, 50]. Denoting the largest node degree in a graph as $d$, the time complexity of GHOST [35] is $\mathcal{O}(d^4)$. The methods above take the graph affinity matrix as input, so their memory complexity in the worst case is $\mathcal{O}(V^4)$. MI-GRAAL [23], HubAlign [19] and NETAL [32] are relatively efficient, with time complexity $\mathcal{O}(VE + V^2 \log V)$, $\mathcal{O}(V^2 \log V)$ and $\mathcal{O}(E^2 + EV \log V)$, respectively. CPD+Emb first learns $D$-dimensional node embeddings [18], and then registers the embeddings by the CPD method [31], whose time complexity is $\mathcal{O}(DV^2)$. The memory complexity of these four methods is $\mathcal{O}(V^2)$. For GW discrepancy-based methods, the GWL+Emb in [49] achieves graph matching and node embedding jointly. It uses the distance matrix of node embeddings and breaks the sparsity of edges, so its time complexity is $\mathcal{O}(V^3)$ and memory complexity is $\mathcal{O}(V^2)$. The time complexity of GWL is $\mathcal{O}(VE)$, but its memory complexity is still $\mathcal{O}(V^2)$ because the $\boldsymbol{L}(\boldsymbol{C}_s, \boldsymbol{C}_t, \boldsymbol{T}^{(n)})$ in (3) is a dense matrix. Our S-GWL combines the recursive mechanism with the regularized proximal gradient method and implements the $\boldsymbol{C}_s \boldsymbol{T}^{(n)} \boldsymbol{C}_t^\top$ in (3) by sparse matrix multiplications. Ideally, we can apply $R = \lfloor \log_K V \rfloor$ recursions. In the $r$-th recursion we calculate $K^r$ barycenter graphs for $K^r$ sub-graph pairs. The sub-graphs in each pair have $\mathcal{O}(\frac{V}{K^r})$ nodes. As a result, we have

**Proposition 3.1.** *Suppose that we have $M$ graphs, each of which has $V$ nodes and $E$ edges. With the help of the recursive $K$-partition mechanism, the time complexity of our S-GWL method is $\mathcal{O}(MK(E + V) \log_K V)$, and its memory complexity is $\mathcal{O}(M(E + VK))$.*

Choosing $K = 2$ and ignoring the number of graphs, we obtain the complexity shown in Table 1. Our S-GWL has lower computational time complexity and memory requirements than many existing

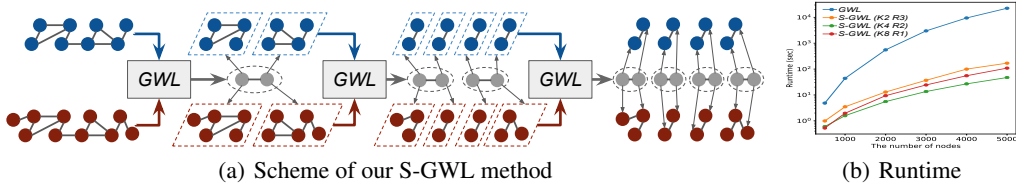

(a) Scheme of our S-GWL method          (b) Runtime

Figure 2: (a) An illustration of S-GWL. (b) Comparisons on runtime.

Table 1: Comparisons for graph matching methods on time and memory complexity.

| | Hungarian | GHOST* | MI-GRAAL | HubAlign | NETAL | CPD+Emb | GWL+Emb | GWL | S-GWL |
|---|---|---|---|---|---|---|---|---|---|
| Time $\mathcal{O}(\cdot)$ | $V^3$ | $d^4$ | $VE+V^2\log V$ | $V^2\log V$ | $E^2+EV\log V$ | $DV^2$ | $V^3$ | $VE$ | $2(E+V)\log V$ |
| Memory $\mathcal{O}(\cdot)$ | $V^4$ | $V^4$ | $V^2$ | $V^2$ | $V^2$ | $V^2$ | $V^2$ | $V^2$ | $E+2V$ |

* $d$ is the largest node degree in a graph.

methods. Figure 2(b) visualizes the runtime of GWL and S-GWL on matching synthetic graphs. The S-GWL methods with different configurations (*i.e.*, the number of partitions $K$ and that of recursions $R$) are consistently faster than GWL. More detailed analysis is provided in Appendix A.3.

## 4   Related Work

**Gromov-Wasserstein learning** GW discrepancy has been applied in many matching problems, *e.g.*, registering 3D objects [28, 29] and matching vocabulary sets between different languages [2]. Focusing on graphs, a fused Gromov-Wasserstein distance is proposed in [44, 43], combining GW discrepancy with Wasserstein discrepancy [46]. The work in [49] further takes node embedding into account, learning the GW discrepancy between two graphs and their node embeddings jointly. The appropriateness of these methods is supported by [11], which proves that GW discrepancy is a pseudometric on measure graphs. Recently, an adversarial learning method based on GW discrepancy is proposed in [9], which jointly trains two generative models in incomparable spaces. The work in [37] further proposes Gromov-Wasserstein barycenters for clustering distributions and interpolating shapes. Currently, GW discrepancy is mainly calculated based on Sinkhorn iterations [41, 15, 5, 37], whose applications to large-scale graphs are challenging because of its high complexity. Our S-GWL method is the first attempt to make GW discrepancy applicable to large-scale graph analysis.

**Graph partitioning and graph matching** Graph partitioning is important for community detection in networks. Many graph partitioning methods have been proposed, such as Metis [21], EdgeBetweenness [16], FastGreedy [12], Label Propagation [38], Louvain [6] and Fluid Community [34]. All of these methods explore the clustering structure of nodes heuristically based on the modularity-maximization principle [16, 12]. Graph matching is important for network alignment [39, 40, 54] and 2D/3D object registration [31, 51, 20, 53]. Traditional methods formulate graph matching as a quadratic assignment problem (QAP) and solve it based on the Hungarian algorithm [17, 33, 51, 50], which are only applicable to small graphs. For large graphs like protein networks, many heuristic methods have been proposed, such as GRAAL [22], IsoRank [40], PISwap [10], MAGNA++ [45], NETAL [32], HubAlign [19], and GHOST [35], which mainly focus on two-graph matching and are sensitive to the noise in graphs. With the help of GW discrepancy, our work establishes a unified framework for graph partitioning and matching, that can be readily extended to multi-graph cases.

## 5   Experiments

The implementation of our S-GWL method can be found at https://github.com/HongtengXu/s-gwl. We compare it with state-of-the-art methods for graph partitioning and matching. All the methods are run on an Intel i7 CPU with 4GB memory. Implementation details and a further set of experimental results are provided in Appendix B.

### 5.1   Graph partitioning

We first verify the performance of the **GWL** framework on graph partitioning, comparing it with the following four baselines: **Metis** [21], **FastGreedy** [12], **Louvain** [6], and **Fluid** Community [34]. We consider synthetic and real-world data. Similar to [52], we compare these methods in terms of adjusted mutual information (AMI) and runtime. Each synthetic graph is a Gaussian random partition graph with $N$ nodes and $K$ clusters. The size of each cluster is drawn from a normal distribution $\mathcal{N}(200, 10)$. The nodes are connected within clusters with probability $p_{\text{in}}$ and between clusters with probability $p_{\text{out}}$. The ratio $\frac{p_{\text{out}}}{p_{\text{in}}}$ indicates the clearness of the clustering structure, and accordingly

Table 2: Comparisons for graph partitioning methods on AMI, time complexity and runtime (second).

| Method | Metis | | FastGreedy | | Louvain | | Fluid | | GWL | |
|---|---|---|---|---|---|---|---|---|---|---|
| Time complexity | $\mathcal{O}(V+E+K\log K)$ | | $\mathcal{O}(VE\log V)$ | | $\mathcal{O}(V\log V)$ | | $\mathcal{O}(E)$ | | $\mathcal{O}((E+V)K)$ | |
| $(N, p_{\text{in}}, p_{\text{out}})$ | AMI | Time | AMI | Time | AMI | Time | AMI | Time | AMI | Time |
| $(4000, 0.2, 0.05)$ | 0.413 | 1.744 | 0.247 | 55.435 | 0.747 | 22.889 | 0.776 | 21.580 | **0.812** | 13.033 |
| $(4000, 0.2, 0.1)$ | 0.009 | 2.340 | 0.064 | 65.441 | 0.574 | 95.114 | 0.577 | 111.043 | **0.590** | 12.740 |
| $(4000, 0.2, 0.15)$ | 0.002 | 3.592 | 0.002 | 80.322 | 0.005 | 290.846 | 0.005 | 203.225 | **0.012** | 12.901 |

Table 3: Comparisons for graph partitioning methods on AMI.

| Method | Metis | | FastGreedy | | Louvain | | Fluid | | GWL | |
|---|---|---|---|---|---|---|---|---|---|---|
| Dataset | Raw | Noisy | Raw | Noisy | Raw | Noisy | Raw | Noisy | Raw | Noisy |
| EU-Email | 0.421 | 0.246 | 0.312 | 0.118 | 0.434 | 0.272 | — | 0.338 | **0.459** | **0.349** |
| Indian-Village | 0.834 | 0.513 | **0.882** | 0.275 | 0.880 | 0.633 | — | 0.401 | 0.857 | **0.664** |

"—": Fluid is inapplicable when the networks have disconnected nodes or sub-graphs.

the difficulty of partitioning. We set $N = 4000$, $p_{\text{in}} = 0.2$, and $p_{\text{out}} \in \{0.05, 0.1, 0.15\}$. Under each configuration $(N, p_{\text{in}}, p_{\text{out}})$, we simulate 10 graphs. For each method, its average performance on these 10 graphs is listed in Table 2. GWL outperforms the alternatives consistently on AMI. Additionally, as shown in Table 2, GWL has time complexity comparable to other methods, especially when the graph is sparse, $e.g.$, $E = \mathcal{O}(V\log V)$. According to the runtime in practice, GWL is faster than most baselines except Metis, likely because Metis is implemented in the C language while GWL and other methods are based on Python.

Table 3 lists the performance of different methods on two real-world datasets. The first dataset is the email network from a large European research institution [25]. The network contains 1,005 nodes and 25,571 edges. The edge $(v_i, v_j)$ in the network mean that person $v_i$ sent person $v_j$ at least one email, and each node in the network belongs to exactly one of 42 departments at the research institute. The second dataset is the interactions among 1,991 villagers in 12 Indian villages [3]. Furthermore, to verify the robustness of GWL to noise, we not only consider the raw data of these two datasets but also create their noisy version by adding 10% more noisy edges between different communities ($i.e.$, departments and villages). Experimental results show that GWL is at least comparable to its competitors on raw data, and it is more robust to noise than other methods.

## 5.2 Graph matching

For two-graph matching, we compare our S-GWL method with the following baselines: **PISwap** [10], **GHOST** [35], **MI-GRAAL** [23], **MAGNA++** [45], **HubAlign** [19], **NETAL** [32], **CPD+Emb** [18, 31], the **GWL** framework based on Algorithm 1, and the **GWL+Emb** in [49]. We test all methods on both synthetic and real-world data. For each method, given the learned correspondence set $\mathcal{S}$ and the ground-truth correspondence set $\mathcal{S}_{real}$, we calculate node correctness as NC $= |\mathcal{S} \cap \mathcal{S}_{real}|/|\mathcal{S}| \times 100\%$. The runtime of each method is recorded as well.

In the synthetic dataset, each source graph $G(\mathcal{V}_s, \mathcal{E}_s)$ obeys a Gaussian random partition model [7] or Barabási-Albert model [4]. For each source graph, we generate a target graph by adding $|\mathcal{V}_s| \times q\%$ noisy nodes and $|\mathcal{E}_s| \times q\%$ noisy edges to the source graph. Figure 1(e) compares our S-GWL with the baselines when $|\mathcal{V}_s| = 2000$ and $q = 5$. For each method, its average node correctness and runtime on matching 10 synthetic graph pairs are plotted. Compared with existing heursitic methods, GW discrepancy-based methods (GWL+Emb, GWL and S-GWL) obtain much higher node correctness. GWL+Emb achieves the highest node correctness, with runtime comparable to many baselines. Our GWL framework does not learn node embeddings when matching graphs, so it is slightly worse than GWL+Emb on node correctness but achieves about 10 times acceleration. Our S-GWL method further accelerates GWL with the help of the recursive mechanism. It obtains high node correctness and makes its runtime comparable to the fastest methods (HubAlign and NETAL).

In addition to graph matching on synthetic data, we also consider two real-world matching tasks. The first task is matching the protein-protein interaction (PPI) network of yeast with its noisy version. The PPI network of yeast contains 1,004 proteins and their 4,920 high-confidence interactions. Its noisy version contains $q\%$ more low-confidence interactions, and $q \in \{5, 10, 15, 20, 25\}$. The dataset is available on https://www3.nd.edu/~cone/MAGNA++/. The second task is matching user accounts in different communication networks. The dataset is available on http://vacommunity.org/VAST+Challenge+2018+MC3, which records the communications among a company's employees. Following the work in [49], we extract 622 employees and their *call-network* and *email-network*.

Table 4: Comparisons for graph matching methods on node correctness (%) and runtime (second).

| Dataset | Yeast 5% noise | | Yeast 15% noise | | Yeast 25% noise | | MC3 sparse | | MC3 dense | |
|---|---|---|---|---|---|---|---|---|---|---|
| Method | NC | Time | NC | Time | NC | Time | NC | Time | NC | Time |
| PISwap | 0.10 | 15.80 | 0.10 | 18.31 | 0.00 | 22.09 | 6.32 | 10.27 | 0.00 | 11.81 |
| GHOST | 11.06 | 25.67 | 0.40 | 30.22 | 0.30 | 35.54 | 21.27 | 17.86 | 0.03 | 22.90 |
| MI-GRAAL | 18.03 | 189.21 | 6.87 | 202.77 | 5.18 | 240.03 | 35.53 | 72.89 | 0.64 | 197.65 |
| MAGNA++ | 48.13 | 603.29 | 25.04 | 630.60 | 13.61 | 624.17 | 7.88 | 425.16 | 0.09 | 447.86 |
| HubAlign | 50.00 | 3.27 | 35.16 | 3.50 | 12.85 | 3.89 | 36.21 | 2.11 | 3.86 | 2.29 |
| NETAL | 6.87 | 1.91 | 0.90 | 2.06 | 1.00 | 2.09 | 36.87 | 1.23 | 1.77 | 1.30 |
| CPD+Emb | 3.59 | 103.22 | 2.09 | 110.19 | 2.00 | 108.62 | 4.35 | 87.54 | 0.48 | 95.68 |
| GWL+Emb | 83.66 | 1340.58 | 66.63 | 1499.20 | 57.97 | 1537.93 | 40.45 | 608.76 | 4.23 | 831.80 |
| GWL | 82.37 | 190.97 | 65.34 | 212.16 | 58.76 | 210.86 | 34.21 | 89.43 | 3.96 | 93.94 |
| S-GWL | 81.08 | 68.58 | 61.85 | 70.06 | 56.27 | 74.64 | 36.92 | 8.39 | 4.03 | 9.01 |

Table 5: Comparisons for multi-graph matching methods on yeast networks.

| Method | 3 graphs | | 4 graphs | | 5 graphs | | 6 graphs | |
|---|---|---|---|---|---|---|---|---|
| | NC@1 | NC@all | NC@1 | NC@all | NC@1 | NC@all | NC@1 | NC@all |
| MultiAlign | 62.97 | 45.19 | — | — | — | — | — | — |
| GWL | **63.84** | **46.22** | **68.73** | **39.14** | 71.61 | 31.57 | 76.49 | 28.39 |
| S-GWL | 60.06 | 43.33 | 68.53 | 38.45 | **73.21** | **33.27** | **76.99** | **29.68** |

For each communication network, we construct a dense version and a sparse one: the dense version keeps all the communications (edges) among the employees, while the sparse version only preserves the communications happening more than $8$ times. We test different methods on $i$) matching yeast's PPI network with its $5\%$, $15\%$ and $25\%$ noisy versions; and $ii$) matching the employee call-network with their email-network in both sparse and dense cases. Table 4 shows the performance of various methods in these two tasks. Similar to the experiments on synthetic data, the GW discrepancy-based methods outperform other methods on node correctness, especially for highly-noisy graphs, and our S-GWL method achieves a good trade-off between accuracy and efficiency.

Given the PPI network of yeast and its 5 noisy versions, we test GWL and S-GWL for multi-graph matching. We consider several existing multi-graph matching methods and find that the methods in [33, 51, 50] are not applicable for the graphs with hundreds of nodes because $i$) their time complexity is at least $\mathcal{O}(V^3)$, and $ii$) they suffer from inadequate memory on our machine (with 4GB memory) because their memory complexity in the worst case is $\mathcal{O}(V^4)$. The IsoRankN in [26] can align multiple PPI networks jointly, but it needs confidence scores of protein pairs as input, which are not available for our dataset. The only applicable baseline we are aware of is the **MultiAlign** in [54]. However, it can only achieve three-graph matching. Table 5 lists the performance of various methods. Given learned correspondence sets, each of which is a set of matched nodes from different graphs, NC@1 represents the percentage of the set containing at least a pair of correctly-matched nodes, and NC@all represents the percentage of the set in which arbitrary two nodes are matched correctly. Both GWL and S-GWL obtain comparable performance to MultiAlign on three-graph matching, and GWL is the best. When the number of graphs increases, NC@1 increases while NC@all decreases for all the methods, and S-GWL becomes even better than GWL.

# 6 Conclusion and Future Work

We have developed a scalable Gromov-Wasserstein learning method, achieving large-scale graph partitioning and matching in a unified framework, with theoretical support. Experiments show that our approach outperforms state-of-the-art methods in many situations. However, it should be noted that our S-GWL method is sensitive to its hyperparameters. Specifically, we observed in our experiments that the $\gamma$ in (3) should be set carefully according to observed graphs. Generally, for large-scale graphs we have to use a large $\gamma$ and solve (3) with many iterations. The $a$ and $b$ in (5) are also significant for the performance of our method. The settings of these hyperparameters and their influences are shown in Appendix B. In the future, we will further study the influence of hyperparameters on the rate of convergence and set the hyperparameters adaptively according to observed data. Additionally, our S-GWL method can decompose a large graph into many independent small graphs, so we plan to further accelerate it by parallel processing and/or distributed learning.

**Acknowledgements** This research was supported in part by DARPA, DOE, NIH, ONR and NSF. We thank Dr. Hongyuan Zha for helpful discussions.

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
