[Supplementary Material · S_GWL_NeurIPS2019-final-main+supp.pdf]

# Scalable Gromov-Wasserstein Learning for Graph Partitioning and Matching

**Hongteng Xu**[1,2]   **Dixin Luo**[2]   **Lawrence Carin**[2]
[1]Infinia ML Inc.   [2]Duke University
{hongteng.xu, dixin.luo, lcarin}@duke.edu

## Abstract

We propose a scalable Gromov-Wasserstein learning (S-GWL) method and establish a novel and theoretically-supported paradigm for large-scale graph analysis. The proposed method is based on the fact that Gromov-Wasserstein discrepancy is a pseudometric on graphs. Given two graphs, the optimal transport associated with their Gromov-Wasserstein discrepancy provides the correspondence between their nodes and achieves graph matching. When one of the graphs has isolated but self-connected nodes ($i.e.$, a disconnected graph), the optimal transport indicates the clustering structure of the other graph and achieves graph partitioning. Using this concept, we extend our method to multi-graph partitioning and matching by learning a Gromov-Wasserstein barycenter graph for multiple observed graphs; the barycenter graph plays the role of the disconnected graph, and since it is learned, so is the clustering. Our method combines a recursive $K$-partition mechanism with a regularized proximal gradient algorithm, whose time complexity is $\mathcal{O}(K(E+V)\log_K V)$ for graphs with $V$ nodes and $E$ edges. To our knowledge, our method is the first attempt to make Gromov-Wasserstein discrepancy applicable to large-scale graph analysis and unify graph partitioning and matching into the same framework. It outperforms state-of-the-art graph partitioning and matching methods, achieving a trade-off between accuracy and efficiency.

## 1   Introduction

Gromov-Wasserstein distance [42, 29] was originally designed for metric-measure spaces, which can measure distances between distributions in a relational way, deriving an optimal transport between the samples in distinct spaces. Recently, the work in [11] proved that this distance can be extended to *Gromov-Wasserstein discrepancy* (GW discrepancy) [37], which defines a pseudometric for graphs. Accordingly, the optimal transport between two graphs indicates the correspondence between their nodes. This work theoretically supports the applications of GW discrepancy to structural data analysis, $e.g.$, 2D/3D object matching [30, 28, 8], molecule analysis [43, 44], network alignment [49], etc. Unfortunately, although GW discrepancy-based methods are attractive theoretically, they are often inapplicable to large-scale graphs, because of high computational complexity. Additionally, these methods are designed for two-graph matching, ignoring the potential of GW discrepancy to other tasks, like graph partitioning and multi-graph matching. As a result, the partitioning and the matching of large-scale graphs still typically rely on heuristic methods [16, 12, 45, 27], whose performance is often sub-optimal, especially in noisy cases.

Focusing on the issues above, we design a scalable Gromov-Wasserstein learning (S-GWL) method and establish a new and unified paradigm for large-scale graph partitioning and matching. As illustrated in Figure 1(a), given two graphs, the optimal transport associated with their Gromov-Wasserstein discrepancy provides the correspondence between their nodes. Similarly, graph partitioning corresponds to calculating the Gromov-Wasserstein discrepancy between an observed graph and a disconnected graph, as shown in Figure 1(b). The optimal transport connects each node of the ob-

(a) Graph matching   (c) Multi-graph matching

(b) Graph partitioning   (d) Multi-graph partitioning

(e) Comparisons on accuracy and efficiency

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

## Footnotes

[1]The memory complexity actually should be $\mathcal{O}(E_s + E_t + V_s V_t)$. Based on the sparsity of edge, we ignore the edge-related terms.

[2]Even if edges are sparse, $E_s$ is often comparable to $V_s K$. Therefore, different from the analysis for Algorithms 3 and 5, here we do not ignore $E_s$.

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

# A Details of Algorithms

## A.1 The GWL framework for different tasks

Based on Algorithms 1 and 2, our GWL framework achieve the graph partitioning and matching tasks in Figures 1(a)-1(d). The schemes of GWL for these tasks are shown in Algorithms 3-6.

---

**Algorithm 3** $\mathcal{S} = \text{GWL-GraphMatching}(G_s, G_t, \gamma)$

---

**Require:** $G_s = G(\mathcal{V}_s, \boldsymbol{C}_s, \boldsymbol{\mu}_s)$, $G_t = G(\mathcal{V}_t, \boldsymbol{C}_t, \boldsymbol{\mu}_t)$, hyperparameter $\gamma$.
1: Initialize correspondence set $\mathcal{S} = \emptyset$.
2: $\widehat{\boldsymbol{T}} = \text{ProxGrad}(G_s, G_t, \gamma)$.
3: **For** $v_i^s \in \mathcal{V}_s$
4:     Find $j = \arg\max_j \widehat{T}_{ij}$, then $\mathcal{S} = \mathcal{S} \cup \{(v_i^s, v_j^t)\}$.
5: **return** $\mathcal{S}$

---

**Algorithm 4** $\{G_k\}_{k=1}^K = \text{GWL-GraphPartitioning}(G, \gamma, K)$

---

**Require:** $G = G(\mathcal{V}, \boldsymbol{C}, \boldsymbol{\mu})$, hyperparameter $\gamma$, the number of clusters $K$.
1: Initialize a node distribution via (6): $\boldsymbol{\mu}_{\text{dc}} = \text{interpolate}_K(\text{sort}(\boldsymbol{\mu}))$
2: Construct a disconnected graph $G_{\text{dc}} = G(\mathcal{V}_{\text{dc}}, \text{diag}(\boldsymbol{\mu}_{\text{dc}}), \boldsymbol{\mu}_{\text{dc}})$, where $\mathcal{V}_{\text{dc}} = \{1, ..., K\}$.
3: $\widehat{\boldsymbol{T}} = \text{ProxGrad}(G, G_{\text{dc}}, \gamma)$.
4: Initialize $\mathcal{V}_k = \emptyset$ for $k = 1, ..., K$.
5: **For** $v_i \in \mathcal{V}$
6:     Find $j = \arg\max_j \widehat{T}_{ij}$, then $\mathcal{V}_j = \mathcal{V}_j \cup \{v_i\}$.
7: **For** $k = 1, ..., K$
8:     Construct a adjacency matrix by selecting rows and columns: $\boldsymbol{C}_k = \boldsymbol{C}(\mathcal{V}_k, \mathcal{V}_k)$.
9:     Construct a node distribution by selecting elements and normalizing them: $\boldsymbol{\mu}_k = \frac{\boldsymbol{\mu}(\mathcal{V}_k)}{\|\boldsymbol{\mu}(\mathcal{V}_k)\|_1}$.
10: **return** $\{G_k = G(\mathcal{V}_k, \boldsymbol{C}_k, \boldsymbol{\mu}_k)\}_{k=1}^K$

---

**Algorithm 5** $\mathcal{S} = \text{GWL-MultiGraphMatching}(\mathcal{G}, \gamma)$

---

**Require:** A graph set $\mathcal{G} = \{G_m = G(\mathcal{V}_m, \boldsymbol{C}_m, \boldsymbol{\mu}_m)\}_{m=1}^M$, hyperparameter $\gamma$
1: Initialize correspondence set $\mathcal{S} = \emptyset$, $K = \min\{|\mathcal{V}_m|\}_{m=1}^M$, $\boldsymbol{\omega} = [\frac{1}{M}, ..., \frac{1}{M}]$.
2: $\{\widehat{\boldsymbol{T}}_m\}_{m=1}^M = \text{GWB}(\{G_m\}_{m=1}^M, \gamma, K, \boldsymbol{\omega})$.
3: **For** $k = 1, ..., K$
4:     $\boldsymbol{s} = \emptyset$
5:     **For** $m = 1, .., M$
6:         Find $i = \arg\max_i \widehat{T}_{ik}^m$, then $\boldsymbol{s} = \boldsymbol{s} \cup \{v_i^m\}$.
7:     $\mathcal{S} = \mathcal{S} \cup \boldsymbol{s}$.
8: **return** $\mathcal{S}$.

---

**Algorithm 6** $\{\mathcal{G}_k\}_{k=1}^K = \text{GWL-MultiGraphPartitioning}(\mathcal{G}, \gamma, K)$

---

**Require:** A graph set $\mathcal{G} = \{G_m = G(\mathcal{V}_m, \boldsymbol{C}_m, \boldsymbol{\mu}_m)\}_{m=1}^M$, hyperparameter $\gamma$, the number of clusters $K$.
1: Initialize $\boldsymbol{\omega} = [\frac{1}{M}, ..., \frac{1}{M}]$.
2: $\{\widehat{\boldsymbol{T}}_m\}_{m=1}^M = \text{GWB}(\{G_m\}_{m=1}^M, \gamma, K, \boldsymbol{\omega})$.
3: Initialize $\mathcal{V}_{k,m} = \emptyset$ for $k = 1, .., K$ and $m = 1, .., M$.
4: **For** $m = 1, .., M$
5:     **For** $v_i^m \in \mathcal{V}_m$
6:         Find $j = \arg\max_j \widehat{T}_{ij}^m$, then $\mathcal{V}_{j,m} = \mathcal{V}_{j,m} \cup \{v_i^m\}$.
7:     **For** $k = 1, ..., K$
8:         $\boldsymbol{C}_{k,m} = \boldsymbol{C}_m(\mathcal{V}_{k,m}, \mathcal{V}_{k,m})$, and $\boldsymbol{\mu}_{k,m} = \frac{\boldsymbol{\mu}_m(\mathcal{V}_{k,m})}{\|\boldsymbol{\mu}(\mathcal{V}_{k,m})\|_1}$.
9: **return** $\{\mathcal{G}_k\}_{k=1}^K$, where $\mathcal{G}_k = \{G_{k,m} = G(\mathcal{V}_{k,m}, \boldsymbol{C}_{k,m}, \boldsymbol{\mu}_{k,m})\}_{m=1}^M$.

---

## A.2 The scheme of S-GWL

Based on Algorithms 3, 5 and 6, we show the scheme of our S-GWL method for (multi-) graph matching in Algorithm 7.

---

**Algorithm 7** $\mathcal{S} = \text{S-GWL}(\mathcal{G}_0, \gamma, K, R)$

---

**Require:** A graph set with $M$ graphs, $i.e.$, $\mathcal{G}_0 = \{G_m = G(\mathcal{V}_m, \boldsymbol{C}_m, \boldsymbol{\mu}_m)\}_{m=1}^{M}$, $\gamma$, the number of partitions $K$ and that of recursions $R$.
1: Initialize correspondence set $\mathcal{S} = \emptyset$.
2: Initialize the root collection of graph sets as $\mathsf{G}_0 = \{\mathcal{G}_0\}$.
3: **For** $r = 1, ..., R$                          $\backslash\backslash$ Recursive $K$-partition mechanism
4:     Initialize $\mathsf{G}_r = \emptyset$.
5:     **For** each graph set $\mathcal{G} \in \mathsf{G}_{r-1}$
6:         $\{\mathcal{G}_k\}_{k=1}^{K} = \text{GWL-MultiGraphPartitioning}(\mathcal{G}, \gamma, K)$.
7:         $\mathsf{G}_r = \mathsf{G}_r \cup \{\mathcal{G}_k\}_{k=1}^{K}$.
8: **For** each graph set $\mathcal{G} \in \mathsf{G}_R$
9:     **If** $M = 2$                                 $\backslash\backslash$ Two-graph matching
10:         $\mathcal{S}_{tmp} = \text{GWL-GraphMatching}(G_s, G_t, \gamma)$, where $\mathcal{G} = \{G_s, G_t\}$.
11:     **Else**                                     $\backslash\backslash$ Multi-graph matching
12:         $\mathcal{S}_{tmp} = \text{GWL-MultiGraphMatching}(\mathcal{G}, \gamma)$.
13:     $\mathcal{S} = \mathcal{S} \cup \mathcal{S}_{tmp}$.
14: **return** $\mathcal{S}$.

---

## A.3 Detailed complexity analysis for GWL and S-GWL

**Algorithms 3 and 5** Suppose that we have a source graph with $V_s$ nodes and $E_s$ edges and a target graph with $V_t$ nodes and $E_t$ edges. The most time- and memory-consuming operation in Algorithm 3 is the $\boldsymbol{C}_s \boldsymbol{T}^{(n)} \boldsymbol{C}_t^{\top}$ in (3). Because $\boldsymbol{C}_s$ is with size $V_s \times V_s$ and $\boldsymbol{C}_t$ is with size $V_t \times V_t$, the computational time complexity of this step in the worst case is $\mathcal{O}(V_s^2 V_t + V_s V_t^2)$ and its memory complexity is $\mathcal{O}(V_s^2 + V_t^2 + V_s V_t)$. Taking advantage of the sparsity of edge, $\boldsymbol{C}_s \boldsymbol{T}^{(n)} \boldsymbol{C}_t^{\top}$ can be implemented by sparse matrix multiplications ($i.e.$, save $\boldsymbol{C}_s$, $\boldsymbol{C}_t$ as "csr" matrix in Python), whose computational time complexity and memory cost can be reduced to $\mathcal{O}(E_s V_t + V_s E_t)$ and $\mathcal{O}(V_s V_t)^1$, respectively. Assuming that these two graphs are with comparable size, we ignore the number of graphs and the subscripts and rewrite the time and memory complexity as $\mathcal{O}(VE)$ and $\mathcal{O}(V^2)$, as shown in the "GWL" column of Table 1.

Algorithm 5 is a natural extension of Algorithm 3 based on GWB. Suppose that we have $M$ graphs. We assume that these graphs and the target barycenter graph are with comparable size. The computational time complexity of Algorithm 5 is $\mathcal{O}(MVE)$ and its memory complexity is $\mathcal{O}(MV^2)$.

**Algorithms 4 and 6** The main difference between Algorithm 4 and Algorithm 3 is that the size of target graph is much smaller than that of source graph, $i.e.$, $K = V_t \ll V_s$ and $K = E_t$, because the target graph is disconnected, whose number of nodes indicates the number of partitions in the source graph. According to the analysis above, the time and memory complexity of Algorithm 4 is $\mathcal{O}(E_s K + V_s K)$ and $\mathcal{O}(E_s + V_s K)^2$. Ignoring the subscripts, we obtain the complexity shown in Table 2.

Similarly, Algorithm 6 is an extension of Algorithm 4 for $M$ graphs, whose time and memory complexity is $\mathcal{O}(MK(E + V))$ and $\mathcal{O}(M(E + VK))$, respectively.

**Algorithm 7** Given $M$ graphs with comparable sizes, each of which has about $V$ nodes and $E$ edges, we can apply $R = \lfloor \log_K V \rfloor$ recursions. In the $r$-th recursion, the $\mathsf{G}_r$ in Algorithm 7) contains $K^r$ sub-graph sets. If we assume that each partitioning operation partition a graph into $K$ sub-graphs with comparable sizes, the $m$-th sub-graph in each set should be with $\mathcal{O}(\frac{V}{K^r})$ nodes and $\mathcal{O}(\frac{E}{K^r})$ edges. For each sub-graph set, we calculate its barycenter graph by Algorithm 6, thus, its time and memory complexity is $\mathcal{O}(MK(\frac{E}{K^r} + \frac{V}{K^r}))$ and $\mathcal{O}(\frac{M}{K^r}(E + VK))$, respectively. At the end of recursion,

Figure 3: Illustrations of the improvements on convergence achieved by our proximal gradient method regularized by node prior (*i.e.*, "prior + proximal" compared with the entropy-based method in [37]) and the vanilla proximal gradient method in [49].

we obtain $K^R$ sub-graph sets. Each sub-graph is very small, with size $\mathcal{O}(\frac{V}{K^R})$. As long as $K^R$ is comparable to $V$, the computations in lines 8-13 of Algorithm 7 can be ignored compared with the computations in the recursions.

In summary, we run $\lfloor \log_K V \rfloor$ recursions, and in the $r$-th recursion we need to calculate $K^r$ barycenter graphs. The overall time complexity of S-GWL is $\mathcal{O}(MK(E+V)\log_K V)$, and its memory complexity is $\mathcal{O}(M(E+VK))$, respectively, as shown in Proposition 3.1. Choosing $K=2$ and ignoring the number of graphs, we obtain the complexity shown in Table 1.

### A.4 Usefulness of node prior

With the help of the prior knowledge of node (*i.e.*, $\boldsymbol{C}_{\text{node}}$), our regularized proximal gradient method can achieve a stable optimal transport with few iterations, whose rate of convergence is faster than the entropy-based method in [37] and the vanilla proximal gradient method in [49]. Figure 3 illustrates the improvements on convergence achieved by our method. Given two synthetic graphs with 1,000 nodes, we calculate their GW discrepancy by different methods. Our method can reach lower GW discrepancy with fewer iterations, and its superiority is consistent with respect to the change of the hyperparameter $\gamma$.

## B More Experimental Results

### B.1 Implementation details

For each baseline, we list its source and language below:

- Graph Partitioning:
  - Metis (C): http://glaros.dtc.umn.edu/gkhome/views/metis
  - FastGreedy (Python): https://networkx.github.io/documentation/networkx-2.2/reference/algorithms/generated/networkx.algorithms.community.modularity_max.greedy_modularity_communities.html#networkx.algorithms.community.modularity_max.greedy_modularity_communities
  - Louvain (Python): https://github.com/taynaud/python-louvain
  - Fluid (Python): https://networkx.github.io/documentation/networkx-2.2/reference/algorithms/generated/networkx.algorithms.community.asyn_fluid.asyn_fluidc.html#networkx.algorithms.community.asyn_fluid.asyn_fluidc
- Graph Matching:
  - PISwap (Python): http://cb.csail.mit.edu/cb/piswap/webserver/
  - GHOST (C): http://www.cs.cmu.edu/~ckingsf/software/ghost/
  - MI-GRAAL (C): http://www0.cs.ucl.ac.uk/staff/natasa/MI-GRAAL/index.html
  - MAGNA++ (C): https://www3.nd.edu/~cone/MAGNA++/

Table 6: The settings of hyperparameters in different experiments.

| Experiments | $\tau$ | $a$ | $b$ | $\gamma$ | $K$ | $R$ |
|---|---|---|---|---|---|---|
| Synthetic partitioning (Table 2) | 0 | 0 | 1 | 1e-2 | — | — |
| EU-Email partitioning (Table 3) | 0 | 0 | 1e-3 | 5e-7 | — | — |
| Indian-Village partitioning (Table 3) | 0 | 5e-1 | 1 | 5e-5 | — | — |
| Synthetic matching (Figure 4) | 1e1 | 0 | 1 | 2e-1 | 2 | 3 |
| Yeast graph matching (Table 4) | 1e3 | 0 | 1 | 2.5e-2 | 2 | 3 |
| MC3 network matching (Table 4) | 1e1 | 1 | 1e-1 | 1e-3 | 2 | 3 |
| Yeast multi-graph matching (Table 5) | 1e3 | 0 | 1 | 2.5e-2 | 8 | 1 |
| Yeast-Human matching (Table 7) | 1 | 0 | 5e-1 | 5e-2 | 2 | 4 |

- HubAlign and NETAL (C): https://ttic.uchicago.edu/~hashemifar/
- CPD+Emb (Python): node2vec is from https://snap.stanford.edu/node2vec/, CPD is from https://github.com/siavashk/pycpd.
- GWL+Emb (Python): https://github.com/HongtengXu/gwl.

All the baselines are tested under their default settings. For our GWL framework and S-GWL method, their hyperparameters are set empirically in different experiments, which are shown in Table 6.

Note that using non-uniform node distributions is important for our method, especiically for the cases involving multi-graph partitioning and matching. When doing multi-graph partitioning, the key step of our S-GWL, the adjacency matrix of the barycenter graph is initialized as a diagonal matrix and its node distribution is estimated by the node distributions of observed graphs. The node distribution based on node degree enhances the consistency of the partitioning across different graphs. For example, given two graphs $G_A$ and $G_B$, we jointly partition them into two subgraph pairs $\{G_A^1, G_B^1\}$ and $\{G_A^2, G_B^2\}$. If we use uniform node distributions, the barycenter will be initialized with uniform node distribution $[0.5, 0.5]^\top$ and adjacency matrix $0.5\boldsymbol{I}_2$, and we may have an identification problem — $G_B^2$ can be finally paired with $G_A^1$.

## B.2 Performance on some challenging cases

Although our GWL framework and S-GWL method perform well in most of our experiments, we find some challenging cases that point out our future research direction.

**Matching Barabási-Albert (BA) graphs** Figure 1(e) shows the averaged matching results in 10 trials. In five of these trials, we match synthetic graphs obeying to Gaussian random partition model. In the remaining five trials, we match synthetic graphs obeying to Barabási-Albert (BA) model. The overall performance shown in Figure 1(e) demonstrates the superiority of our S-GWL method. This outstanding result is mainly contributed by the experiments on Gaussian partition graphs. Specifically, when matching Gaussian partition graphs, all the GW discrepancy-based methods achieves very high node correctness, and the speed of our method is almost the same with the fastest HubAlign method, as shown in Figure 4(a). When it comes to BA graphs, Figure 4(b) indicates that although GW discrepancy-based methods still outperform many baselines, there is a gap between them and the state-of-the-art methods in the aspect of node correctness.

Additionally, the BA graphs also have a negative influence on our recursive mechanism. For Gaussian partition graphs, it is relatively easy to partition them into several sub-graphs with comparable size. In such a situation, the power of our recursive mechanism can be maximized, which helps us achieve over 100 times acceleration. However, for BA graphs, the sub-graphs we get are often with incomparable size. The largest sub-graph decides the runtime of our S-GWL method. As a result, our S-GWL method only achieves about 10~20 times acceleration.

Currently, we are making efforts to improve the performance and the speed of our method on BA graphs. To solve this problem, we may need to use some node information, $e.g.$, introducing node embedding into our S-GWL method.

**Matching incomparable graphs** The second challenging case is matching incomparable graphs. This case is common in the field of bioinformatics, $e.g.$, matching the PPI networks from different species. When the networks are with incomparable size, the performance of GW discrepancy-based methods degrades. For example, in Table 7, we match the PPI network of yeast to that of human. This yeast network has 2,340 proteins (nodes), while the human network has 9,141 proteins. Because the ground truth correspondence between these proteins is unknown, we use edge correctness to evaluate

(a) Gaussian Partition: Accuracy v.s. efficiency

(b) Barabási-Albert: Accuracy v.s. efficiency

(c) Gaussian Partition: Acceleration

(d) Barabási-Albert: Acceleration

Figure 4: The performance of our method on different kinds of graphs. (a, b) For each method, its standard deviation of node correctness and that of runtime are shown as well.

Table 7: Comparisons for graph matching methods on edge correctness (%).

| Method | IsoRank | PISwap | MI-GRAAL | GHOST | NETAL | HubAlign | GWL | S-GWL |
|---|---|---|---|---|---|---|---|---|
| Yeast↔Human | 2.12 | 2.16 | 13.87 | 17.04 | 28.65 | 21.59 | 19.56 | 18.89 |

The results of baselines are from [19].

our method. Specifically, edge correctness calculates the percentage of yeast's edges appearing in the human network.

Experimental results show that both GWL and S-GWL outperform most of their competitors except HubAlign and NETAL. The main reason for this phenomenon, in our opinion, is because the constraint of optimal transport. The constraint $T \in \Pi(\boldsymbol{\mu}_s, \boldsymbol{\mu}_t)$ implies that each node in the target graph is assigned to a source node with a probability as long as its probability in $\boldsymbol{\mu}_t$ is nonzero. When the number of target nodes is much larger than that of source nodes, the real correspondence will be oversmoothed because each source node transports to too many target nodes. To overcome this issue, we need to propose a preprocess to remove potentially-useless nodes from the large graph, which is another future work for us.