[Reviews · NeurIPS 2019]

Reviewer 1



In this paper, the authors propose a framework for scalable graph partitioning and matching. They use the pseudometric Gromov-Wasserstein in a scalable setup by utilizing a recursive mechanism. They provide detailed analysis of the complexity of the method. They justify the design of this method and they cover the related work in depth. In the detailed experimental setup, they mention all the comparison methods that are being used, the real and synthetic data, as well as the metrics that those methods will be evaluated. In their experiments for the graph partitioning, they focus both on analyzing the complexity/runtime for their method and the comparison methods, and they provide detailed results on a variant of mutual information metric. For the graph matching task, again the authors provide an analysis for node correctness metric and runtime for their method and the comparison methods. Overall, the paper is about a problem interesting for the NeurIPS community and the authors propose a novel setting to use the Gromov-Wasserstein distance in the graph matching and graph partitioning tasks in a scalable way. The paper is well-written and each claim is well supported by the authors comments', proofs or references. The reviewer has only concerns regarding the experiments and the metrics used in the analysis (see below for metric recommendations).

Reviewer 2



=============== Post-response Update: I thank the authors for their response. As I pointed out in my original review, I think this is an interesting (if somewhat limited in novelty) work, therefore, I maintain my score, and recommend acceptance on the understanding that: 1) The additional results and modifications mentioned in the rebuttal are included in the final version (in particular, details about the node measure). 2) The redundancy pointed out by R3 is discussed in the final version. 3) The "initialization" strategies section is updated and clarified to reflect the additional information provided in the rebuttal =============== Summary: This paper investigates the use of the Gromov-Wasserstein (GW) distance for large-scale graph partitioning and matching. The GW distance produces as a byproduct of its computation a transportation coupling, which can be used to infer node-to-node correspondences between graphs. In addition, the optimal transport framework has the appealing property of generalizing to multi-distribution comparisons thorough barycenters, which is exploited in this work to yield joint multi-graph approaches to partitioning and matching. Instead of a the more traditional projected gradient descent approach, the authors rely on a regularized proximal gradient method to compute the GW distance and barycenters. In order to scale up to large graphs, they propose a recursive divide-and-conquer approach. Various experiments on benchmark graph/network partitioning and matching tasks are performed, showing that the proposed method compares favorably (both in terms of accuracy and runtime) to various popular baselines. Strengths: - Strong theoretical foundations (the Gromov-Wasserstein distance) to a task often approach with heuristic methods - Superbly written paper: clear and concise argumentation, easy to follow, and a pleasure to read - The thorough experimental results, which show that the proposed approach is effective and efficient in practice - Rigorous and comprehensive review of computational complexities of the proposed + alternative methods Weaknesses: - Limited novelty of methods / theory Major Comments/Questions: 1. Novelty/Contributions. While GW has been used for graph matching repeatedly in previous work (albeit for small tasks - see below), I am not aware of other work that uses it for graph partitioning, so I would consider this an important contribution of this paper. It should be noted that most of the individual components used in this work are not novel (the GW itself, its application to graph matching, the proximal gradient method). However, I see consider its main contribution combining those components in a coherent and practical way, and producing as a consequence a promising and well-founded approach to two important tasks. 2. The Scalable method. While recursive partitioning methods are a common fixture of discrete optimization (and thus its use here provides limited novelty), it is satisfactorily applied in this case, and it seems to work well. My main concern/question about this component its is robustness/reliability. Recursive partitioning methods are prone to be unforgiving: I a wrong decision is made in the early stages, it can have disastrous consequences downstream as there is no possibility to revise poor early decisions. This seems to be the case here, so I would be interested to see if and whether the authors observed those catastrophic early mistakes in their experiments, and whether a best-of-k version of their method (e.g., like beam search for sequence decoding) would be able to lessen these. 3. Adjacency matrices. The paper continuously refers to the C matrices as adjacency matrices, yet they are defined over the reals (ie., not binary, as adjacency matrices usually are). I assume they are using soft edges or distances for the entries of these matrices, and that's why they are continuous, but the authors should clarify this. Also on this note, I would have liked to see some intuitive explanation of Eq (4), e.g., I understand it as a soft-averaged (barycenter-weighted) extrapolated similarity matrix. I would suggest the authors to discuss its interpretation, even if briefly. 4. The paper mentions node measures being estimated from node degrees (e.g. L58). How exactly is this done? 5. Initialization. I might be missing something, but I just don't see how the approaches described in the "Optimal Transports" and "Barycenter Graphs" paragraphs are initialization schemes. In particular, the former looks like a regularization scheme applied to every iteration. Could the authors please provide more details about these? 6. Complexity Analysis. I appreciate the thorough yet concise complexity analysis of related methods. This is sadly less and less common in ML papers, so it's encouraging to see it here. One comment: I would suggest reminding the reader what d is in Table 1. 7. Results. I have various minor comments about the experimental results: * The adjusted mutual information should probably be spelled out explicitly (perhaps in the Appendix). I suspect many readers are not familiar with it or don't remember its exact form (I didn't). * Fig 1(e) would be more meaningful with two-sided error bars * What is q%|V| in L273? Is this a product? * Why are there no results for MultiAlign for > 3 graphs? Was it because of timeout? Please mention this in the paper. * NC@1 and NC@all could be better explained - it took me a while to understand what was meant by these Minor Comments/Typos: - L83. "discrepancy" repetition - L107. Two arbitrary nodes - L138. Convergence is not properly linear, but nearly-linear - a detail, yes, but an important one.

Reviewer 3



While I think that the idea definitely worth it, I have some doubts about the fact that the paper is ready for publication. Indeed, it raises some questions that should be treated. Here is a list below. 1) The node distribution is set as the normalized node degree (as in [48]). This choice should be discussed in more details as it is not obvious and may be redundant with matrix C. Why a node with more connections should have a greater weight than others? Why the adjacency matrix not enough for enforcing nodes with similar node degree to be matched? Anyway, this choice deserves a detailed discussion in the paper. 2) The method is applicable to non-attributed graphs (this should be mentioned in the paper). Nevertheless, in section 3.1, authors provide a extra term in the GW formulation, C_node, that involves the differences between the 2 node distributions. The formulation then seems to come down to a GW term + a W term as in the fused Gromov-Wassertein method in [43]. Is this correct? If not, the differences should be highlighted. In addition, would it be possible to consider an other C_node matrix that would involve node labels? 3) Authors consider an entropic version of the GW distance, allowing a faster resolution of the problem. Nevertheless, at least for the graph partitioning problem, considering a non-regularized problem seems to be more intuitive: a node is matched to only one (in most cases) node of the disconnected graph. This is illustrated in figure 1b), where we would have preferred to see the first 3 nodes matching to one node and the 3 others to an other one, instead of having 2 nodes with splitted mass. Authors should justify this choice in more details. 4) the multi-graph partitioning scheme is unclear to me. Does it consists in i) first estimating the barycenter of all the graphs ii) then partitioning the barycenter? In figure 1d), it is not clear what does the transport matrices represent (transport to the barycenter or to the disconnected nodes?). The motivation behind the multi-graph partitioning should also be better explained. 5) A scalable algorithm is given in section 3.2. A discussion about how the results are close of the original solution, and in which cases it can be/should not be used is needed. Indeed, in the experiments, it leads to degraded performances, and this behavior should be better understood. 6) Algorithm in section 3.1 seems to be a direct extension of the algorithm provided in [48]. Originality of the algorithm should be better highlighted. Minor comments: - page 2 "we propose a GW learning framework to unify these two problems": the method proposed to solve these problems is the same but the two problems are definitely different. - regarding the density \mu: what happens if the graph contains isolated nodes? Are they discarded? - page 3 "the maximum in each row indicates the cluster of a node": what happens if some quantities are equal, as it seems to be in fig. 1b)? - page 3: the derivation of the node distribution \mu_dc is probably the most important quantity to be set and its computation details should not appear only in the appendix - for the graph partitioning problem, how do you choose the K value? - in several parts of the paper, assumption that the observed graphs have comparable size is made. Is this a reasonable assumption? **** UPDATE AFTER REBUTTAL**** Thanks for your feedback that I read carefully. It adresses some of my concerns (points 3-4-5-6). I believe that the choice of the node distribution, the cost matrix and C_node should be discussed in more details (all of these seem somehow redundant and some insight about how to set "good" definitions should be added). As such, I modified my score and set it to 6.

[Author Response · NeurIPS 2019]

We thank all the reviewers for their constructive feedback. Below we provide specific responses to each reviewer.

**To Reviewer #1**

We will re-organize Section 2. The metrics like top-$K$ precision are applicable to S-GWL. When matching MC3 sparse
nets, the top-10 precision of S-GWL is 55.26% (higher than its baselines). We will add more results in the paper.

**To Reviewer #2**

**1.** *Our main contribution* is proposing a GWL framework with a non-trivial scalable algorithm for graph matching and
partitioning. In the following response 2, we further highlight our important improvements ignored by existing work.

**2.** *The Method* In Fig.1(e), Tables 4 and 5, S-GWL can be slightly worse than GWL on node correctness. It means that
the recursive partitioning propagates some errors. However, we didn't observe catastrophic mistakes. One reason is that
we just applied 1∼4 recursive steps. Moreover, we suppressed this risk by setting the node distribution of each graph as
normalized node degree, and $i$) considering a regularizer based on node distributions (line 149-155); $ii$) initializing the
node distribution of barycenter graph as the average of the sorted distributions (Eq.(5)). These two improvements make
our S-GWL match the nodes in a graph to those with comparable degrees in another graph. The regularizer improves
the convergence of our algorithm greatly (Fig.3 in Appendix). We tried different $K$'s and the optimal settings are in the
Appendix (Table 6). For different $K$'s, the fluctuations of node correction are $\pm 3.2\%$ in our experiments.

**3.** *Adjacency Matrices* For weighted graphs (the MC3 nets), the adjacency matrices are continuous. For unweighted
graphs (the protein nets), the adjacency matrices are binary. The barycenter is an average of the observed graphs aligned
by their optimal transports. The matrix $\bar{C}$ in Eq.(4) is a "soft" adjacency matrix of the barycenter. Its elements reflect
the confidence of the edges between the corresponding nodes. We will add its interpretation in the revised paper.

**4.** *The Node Distribution* $\boldsymbol{\mu} = \frac{1}{\|\boldsymbol{u}\|_1}\boldsymbol{u}$, where $\boldsymbol{u} = [u_i] \in \mathbb{R}^N$, $u_i$ is the number of edges containing $i$ (node degree).

**5.** *Initialization* Given two graphs, we initialize their optimal transport as the inner product of their node distributions
$\boldsymbol{\mu}_s \boldsymbol{\mu}_t^\top$. For barycenter graph, we initialize its node distribution $\bar{\boldsymbol{\mu}}$ by Eq.(5) and its adjacency matrix as diag($\bar{\boldsymbol{\mu}}$).

**6.** *Complexity* We have defined $d$ in line 188, and we will add a footnote below Table 1 to emphasize it.

**7.** *Results* We will $i$) define AMI; $ii$) add two-sided error bars in Fig.1(e); $iii$) use "$|\mathcal{V}| \times q\%$" in L273; $iv$) explain
NC@1 and NC@All clearly; $v$) fix typos. Given 3 graphs $G_A$, $G_B$, $G_C$, MultiAlign [53] learns the correspondence
$\boldsymbol{T}_{A\to B}, \boldsymbol{T}_{B\to C}, \boldsymbol{T}_{A\to C}$ with a constraint $\boldsymbol{T}_{A\to C} = \boldsymbol{T}_{A\to B}\boldsymbol{T}_{B\to C}$. It does not consider the case of >3 graphs.

**To Reviewer #3**

**1.** We set the node distribution as normalized node degree mainly because of the following reason. When doing
multi-graph partitioning, the key step of our S-GWL, the adjacency matrix of the barycenter graph is initialized as
a diagonal matrix and its node distribution is estimated by the node distributions of observed graphs (*i.e.*, Eq.(5)).
The node distribution based on node degree enhances the consistency of the partitioning across different graphs. For
example, given two graphs $G_A$ and $G_B$, we jointly partition them into two subgraph pairs $\{G_A^1, G_B^1\}$ and $\{G_A^2, G_B^2\}$.
If we use uniform node distributions, the barycenter will be initialized with uniform node distribution $[0.5, 0.5]^\top$ and
adjacency matrix $0.5\boldsymbol{I}_2$, and we may have an identification problem — $G_B^2$ can be finally paired with $G_A^1$.

**2.** We can view the node distribution as 1D features of nodes, then introducing the proposed regularizer indeed leads to
the formulation like the fused GW distance [42, 43]. Given more features, we can use them to compute the $\boldsymbol{C}_{node}$ in
the regularizer. We will highlight this point, but as you mentioned our S-GWL is applicable to non-attributed graphs.

**3.** We tried to partition graphs without the entropy regularizer as you suggested. It achieves at most comparable AMI
but spends more time. Actually, our fuzzy partitioning does not have to be a disadvantage — the nodes with split mass
may indicate the boundary of partitioning, and their mass reflects their closeness to different communities.

**4.** As shown in response 1, our S-GWL jointly partitions $M$ graphs $\{G_m\}_{m=1}^M$ into $K$ subgraph groups $\{\{G_m^k\}_{m=1}^M\}_{k=1}^K$,
such that we can achieve large-scale graph matching by solving $K$ small matching problems. In Fig.1(d), the transport
between each observed graph $G_m$ and the barycenter graph indicates how to partition $G_m$ to $\{G_m^k\}_{k=1}^K$.

**5.** As reviewer #2 mentioned, the recursive partitioning in S-GWL may propagate matching errors. However, in our
experiments the degradation on matching accuracy is often tolerable (1∼2% generally), which means that the matching
results does not change a lot compared with that of the GWL without recursive partitioning. Moreover, the acceleration
achieved by S-GWL is clear. Additionally, as we highlighted in response 2 to reviewer #2, we make improvements on
the algorithm to avoid catastrophic error propagation.

**6.** The proximal gradient method [48] indeed plays an important role in this work. However, it should be noted that to
achieve a scalable GWL method, we improve it by 1) introducing a node-based regularizer, and 2) plugging it into the
computation of barycenter graph. The improved algorithm is further combined with a recursive mechanism.

**7.** For your minor comments: $i$) We will polish our writing in the revised paper. $ii$) The graphs in our experiments do
not have isolated nodes, and we ignore such nodes in practice. $iii$) Currently, we assign the node with equally-splitted
mass to a cluster randomly. This risk also appears in traditional clustering methods like K-means and GMM. According
to our experiments, it does not affect the superiority of our method. $iv$) The computation of node distribution is in the
response 4 to reviewer #2, which will be given in the revised paper. $v$) For graph partitioning, the $K$ is predefined as in
Metis [21]. $vi$) Assuming that the graphs have comparable size is just for the convenience of complexity analysis.

[Meta-Review · NeurIPS 2019]

The reviewers all reached consensus to accept this work --- congratulations! Some requests in the revision: ** address redundancy in the Cost Matrix and C_node ** improve clarity, in particular the missing discussion of the node measure ** make sure that reading this paper does not require detailed knowledge of [48]